# OPPONENT MODELING BASED ON SUBGOAL INFERENCE

## ABSTRACT

When an agent is in a multi-agent environment, it may face previously unseen opponents, and it is a challenge to cooperate with other agents to accomplish the task together or to maximize its own rewards. Most opponent modeling methods deal with the non-stationarity caused by unknown opponent policies via predicting the opponent's actions. However, focusing on the opponent's action is shortsighted, which also constrains the adaptability to unknown opponents in complex tasks. In this paper, we propose *opponent modeling based on subgoal inference*, which infers the opponent's subgoals through historical trajectories. As subgoals are likely to be shared by different opponent policies, predicting subgoals can yield better generalization to unknown opponents. Additionally, we design two subgoal selection modes for cooperative games and general-sum games respectively. Empirically, we show that our method achieves more effective adaptation than existing methods in a variety of complex tasks.

## 1 INTRODUCTION

Reinforcement learning (RL) has achieved remarkable success in games involving multiple agents, such as AlphaGo (Silver et al., 2016), OpenAI Five (OpenAI, 2018), and AlphaStar (Vinyals et al., 2019). The non-stationarity of multi-agent environments has brought many difficulties to problem-solving, and this has always been the case. In cooperative scenarios, many multi-agent reinforcement learning (MARL) methods (Lowe et al., 2017; Sunehag et al., 2017; Rashid et al., 2020; Son et al., 2019) aim to bridge the information gap between agents by training agents in a centralized manner, called centralized training with decentralized execution, enabling agents to work together seamlessly to accomplish cooperative tasks. Alternatively, fully decentralized methods (Jiang & Lu, 2022; Su & Lu, 2022) seek to break free from the constraints of

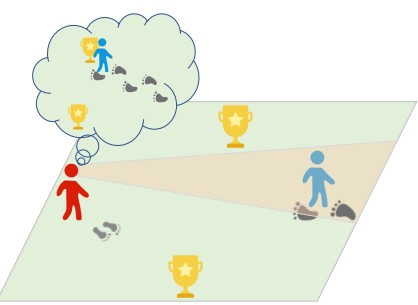

Figure 1: Infer the goal of others

centralized training, allowing agents to reach collaboration in a simpler and decentralized manner. In competitive scenarios, NFSP (Heinrich & Silver, 2016), PSRO (Lanctot et al., 2017), and DeepNash (Perolat et al., 2022) employ self-play to train agents for equilibrium strategies, allowing agents to adapt and improve their policy. By considering how the agent affects the expected learning progress of other agents, LOLA (Foerster et al., 2017) and COLA (Willi et al., 2022) apply opponent shaping to this setting. In these methods, all agents are jointly trained in the same scenario.

Autonomous agents, different from those jointly trained, can act autonomously in complex and dynamic environments, sense the influence of the environment and other agents, and accomplish their own goals or tasks. Such agents can analyze the behavior of opponents[1] by building models that make predictions about some core properties of the agents being modeled, such as their actions, goals, and beliefs, in a method called *opponent modeling* (Albrecht & Stone, 2018). By modeling the intentions and policies of other agents, the training process of the agent might be stabilized (Papoudakis et al., 2019). Many studies rely on predicting the actions (He et al., 2016; Hong et al.,

---

[1]We call any agent other than the autonomous agent itself "opponent," whether it is a teammate or rival.

2018; Grover et al., 2018; Papoudakis & Albrecht, 2020), goals (Raileanu et al., 2018), and returns (Tacchetti et al., 2018) of opponents during training. The autonomous agent adapts to different or unseen opponents by using the predictions or representations that are produced by the relevant modules. However, in some scenarios, opponents may continuously learn during interaction. Meta-MAPG (Kim et al., 2021) combines Meta-PG(Al-Shedivat et al., 2017) and LOLA, and focuses on the problem of the non-stationary environment caused by the continuous learning of opponents. MBOM (Yu et al., 2022) simultaneously targets a variety of adversaries, fixed policy, or continuous learning, by modeling the possible policies that an opponent may form, combined with Bayesian inference to generate an opponent's imagined policy. Some methods focus on figuring out the opponent's goal, *e.g.*, ToMnet (Rabinowitz et al., 2018) and SOM (Raileanu et al., 2018). SOM infers the opponent's goal through its own policy, in other words, "what would I do if I were the opponent?" LIAM (Papoudakis et al., 2021; Papoudakis & Albrecht, 2020) builds the opponent's policy from its own partial observations and uses it to anticipate the opponent's actions and make decisions. GSCU (Fu et al., 2022) chooses online between a real-time greedy strategy and a fixed conservative strategy through Bayesian belief in competitive environments. The greedy strategy is conditioned RL, while the conservative strategy is a bandit algorithm.

Although a lot of the existing methods concentrate on modeling the opponent's actions, such an approach is short-sighted, pedantical, and highly complex. Generally, modeling an opponent's actions is just predicting what it will do at the next step. Intuitively, it is more beneficial for the agent to make decisions if it knows the situation of the opponent several steps ahead. Predicting the actions over a few steps is not elegant. For example, to reach the goal point of $(2, 2)$, an opponent moves from $(0, 0)$ following the action sequence $<\uparrow, \uparrow, \rightarrow, \rightarrow>$ by four steps (Cartesian coordinates). There are also 5 other action sequences, *i.e.*, $<\uparrow, \rightarrow, \uparrow, \rightarrow>, <\uparrow, \rightarrow, \rightarrow, \uparrow>, <\rightarrow, \uparrow, \uparrow, \rightarrow>$ , $<\rightarrow, \uparrow, \rightarrow, \uparrow>, <\rightarrow, \rightarrow, \uparrow, \uparrow>$, that can lead to the same goal. Obviously, the complexity of the action sequence is much higher than the goal itself. Other methods that claim to predict the opponent's goal (Rabinowitz et al., 2018; Raileanu et al., 2018), without explicitly making a connection to the opponent's goal or just predicting the goal at the next step, are essentially as shortsighted as modeling actions.

Inspired by the fact that humans can predict the opponent's goal by observing the opponent's actions for several steps as illustrated in Figure 1, in this paper, we propose ***O**pponent **M**odeling based on sub**G**oals inference* (**OMG**), which uses variational inference to predict the opponent's future subgoals from historical trajectories. The trajectory of an opponent's policy consists of a set of subgoals, and the trajectories of different policies may contain the same subgoal. This combinatorial property of the subgoals facilitates the generalization of the agent to unseen opponents' policies. Moreover, we design two manners for selecting subgoals, which are applied to cooperative games and general sum games, respectively. Empirically, OMG outperforms existing opponent modeling methods in a variety of complex multi-agent environments, demonstrating the superiority of inferring subgoals over predicting actions.

## 2 RELATED WORK

**Opponent modeling.** Opponent modeling plays a crucial role in enhancing the robustness and stability of reinforcement learning (Papoudakis et al., 2019). Given the presence of diverse opponent policies in multi-agent environments, the autonomous agent faces a significant challenge in learning resilient policies. When an agent perceives an opponent as part of the environment, the resulting environment becomes inherently unstable and intricate. To address this challenge, one straightforward method involves equipping the agent with the ability to incorporate information about its opponent, including aspects like the opponent's behavior, goals, and beliefs (Albrecht & Stone, 2018), *i.e.*, opponent modeling. It gives the agent a deeper insight and prediction ability about the opponent's policy. Thus, the autonomous agent views the environment as less unstable and can simply use single-agent reinforcement learning methods.

A common approach to modeling the policy of an opponent is predicting the opponent's actions. DRON (He et al., 2016) and DPIQN (Hong et al., 2018) extend DQN (Mnih et al., 2015) by adding another network that estimates the opponents' actions from the observations. The DQN uses the hidden layer of this network to improve its policy. Variational auto-encoders can also be used to model the opponent's policy (Papoudakis & Albrecht, 2020), which results in probabilistic repre-

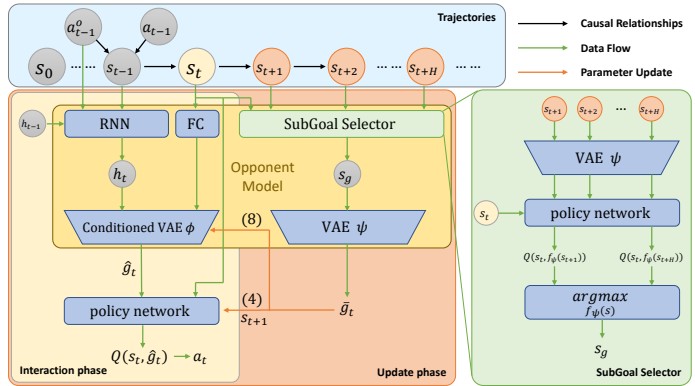

Figure 2: Diagram of OMG architecture. In the interaction phase, OMG deduces subgoals from historical trajectories to enhance decision-making. In the update phase, OMG employs the subgoal selector to choose the state among those within the next few steps as the subgoal.

sentations instead of fixed vectors. PR2 (Wen et al., 2019), MBOM (Yu et al., 2022), and TP-MCTS (Weil et al., 2023) combine the idea of recursive reasoning, nested form as "the agent believes [that the opponent believes (that the agent believes ...)]", based on modeling the action of the opponent. Some works focus on modeling beliefs. Zintgraf et al. (2021) combined the sequential and hierarchical variational auto-encoders to construct a belief inference model using meta-learning, for belief inference. Zhang et al. (2023) introduced landmarks into the behavior model and improve the model by the action sequence of the opponents, so as to recognize and compare the opponent's intention.

Another key aspect of opponent modeling is to infer the opponent's goal. Baker et al. (2009) formulated the goal recognition as a Markov decision process (MDP) and calculate the posterior probability of the goal by Bayes rule based on a prior goal library. ToMnet (Rabinowitz et al., 2018) aims to give the agent a human-like Theory of Mind. It uses three networks to infer the agent's goal and action from previous and present information. SOM (Raileanu et al., 2018) implements the Theory of Mind with a goal library from a different perspective. SOM uses its own policy, the opponent's observation, and the opponent's action to work backward to learn the opponent's goal distribution by gradient ascent. These methods either require a prior goal library or infer implicit "goals" that are not supervised by ground truth goals.

**Goal-conditioned RL.** Goal-conditioned reinforcement learning is an extension of the single-agent algorithm. Most works focus on learning a goal-conditioned policy, where the goals are usually predefined (Plappert et al., 2018; Zhu et al., 2021). Some works consider acquiring subgoals automatically to accelerate learning. Paul et al. (2019) proposed a method that uses expert trajectories to generate subgoals, while (Chane-Sane et al., 2021) proposed to incorporate imaginary subgoals into policy learning to facilitate learning complex tasks, where subgoals are measured by value functions. Unlike existing goal-conditioned RL methods, we aim to infer the subgoal of the opponent and condition the agent policy on the inferred subgoal.

## 3 METHOD

### 3.1 PRELIMINARIES

In general, we consider an $n$-agent stochastic game $\mathcal{M} = (\mathcal{S}, \mathcal{A}^1, \ldots, \mathcal{A}^n, \mathcal{P}, \mathcal{R}^1, \ldots, \mathcal{R}^n, \gamma)$, where $\mathcal{S}$ is the state space, $\mathcal{A}^i$ is the action space of agent $i \in [1, \ldots, n]$, $\mathcal{A} = \prod_{i=1}^{n} \mathcal{A}^i$ is the joint action space of agents, $\mathcal{P} : \mathcal{S} \times \mathcal{A} \times \mathcal{S} \to [0, 1]$ is a transition function, $\mathcal{R}^i : \mathcal{S} \times \mathcal{A} \to \mathbb{R}$ is the reward function of agent $i$, and $\gamma$ is the discount factor. The policy of agent $i$ is $\pi^i$, and the joint policy of other agents is $\pi^o(a^o|s) = \prod_{j \neq i} \pi^j(a^j|s)$, where $a^o$ is the joint action except agent $i$. All agents interact with the environment simultaneously without communication. The historical trajectory is available, *i.e.*, for agent $i$ at timestep $t$, $\tau_t = \{s_0, a_0^i, a_0^o, \ldots, s_{t-1}, a_{t-1}^i, a_{t-1}^o\}$ is observable.

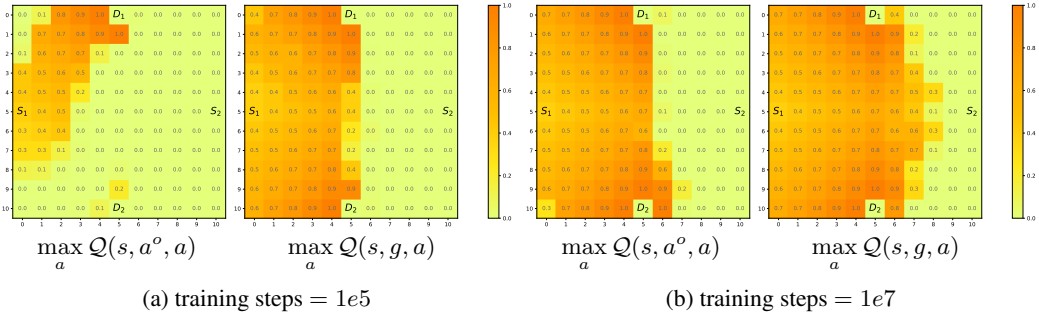

$$\max_a \mathcal{Q}(s, a^o, a) \qquad \max_a \mathcal{Q}(s, g, a) \qquad\qquad \max_a \mathcal{Q}(s, a^o, a) \qquad \max_a \mathcal{Q}(s, g, a)$$

(a) training steps $= 1e5$ \qquad\qquad (b) training steps $= 1e7$

Figure 3: Learned Q-values using tabular Q-learning in an $11 \times 11$ gridworld. The agent and the opponent start from the $S_1$ and $S_2$, respectively. The two reward points are $D_1$ and $D_2$, and the reward will only be given to the agent who arrives first. The opponent executes one of policies $\pi_1^o$ and $\pi_2^o$, which target $D_1$ and $D_2$, respectively.

The goal of the agent $i$ is to maximize its expected cumulative discount rewards:

$$\mathop{\mathbb{E}}_{\substack{s_{t+1}\sim\mathcal{P}(\cdot|s_t,a_t^i,a_t^o), \\ a\sim\pi^i(\cdot|s_t),a_t^o\sim\pi^o(\cdot|s_t)}} \left[\sum_{t=0}^{\infty} \gamma^t \mathcal{R}^i(s_t, a_t^i, a_t^o)\right]. \tag{1}$$

For convenience, the learning agent treats all other agents as a joint opponent with the joint action $a^o \sim \pi^o(\cdot|s)$ and reward $r^o$. The action and reward of the learning agent are respectively denoted as $a \sim \pi(\cdot|s)$ and $r$ for notation simplicity.

An agent treats other agents as part of the environment and ignores the non-stationarity posed by the change of other agents' policies as independent Q-learning (Tampuu et al., 2017; Tan, 1993). Its policy is updated by:

$$\mathcal{Q}(s_t, a_t) = \mathbb{E}_{\mathcal{P}(s_{t+1}|s_t,a^o,a)}[r + \gamma \max_a \mathcal{Q}(s_{t+1}, a)], \tag{2}$$

where $\mathcal{Q}$ is Q-network. Opponent modeling typically predicts the actions of other agents to address the non-stationary problem. The opponent model uses historical trajectory as input to predict $\tilde{a}^o \sim \tilde{\pi}(\cdot|\tau)$, where $\tilde{a}^o$ is the estimate of $a^o$. Its policy is updated as:

$$\mathcal{Q}(s_t, \tilde{a}_t^o, a_t) = \mathbb{E}_{\mathcal{P}(s_{t+1}|s_t,a^o,a)}[r + \gamma \max_a \mathcal{Q}(s_{t+1}, \tilde{a}_{t+1}^o, a)]. \tag{3}$$

Note that we cast our discussion here to Q-learning. All can be similarly applied to other RL methods, such as PPO (Schulman et al., 2017).

## 3.2 POLICY UPDATE WITH OPPONENT'S SUBGOALS

The opponent's subgoal is the representation of the state that the opponent may have in the future based on the opponent's policy. Like "All roads lead to Rome", the opponent may perform different sequences of actions but eventually reach the same state. Instead of focusing on the details of each of the opponent's actions, the agent should focus on the state the opponent wants to reach.

The opponent's subgoal distribution probability is based on the opponent's action sequence, that is, the opponent's policy, but its sample space is still the representation of the state. Here we decouple the subgoal from the opponent's policy and just consider decision-making problems conditioned on the opponent's subgoal. Formally, we transform the original stochastic game $\mathcal{M}$ into a state-augmented MDP, defined by $\mathcal{M}_\mathcal{G} = (\mathcal{S}, \mathcal{G}, \mathcal{A}^i, \mathcal{P}, \mathcal{R}^i, \gamma)$, where $\mathcal{G}$ is the subgoal space. Since $\mathcal{G}$ is a representation of future states the opponent may go, $|\mathcal{G}|$ is finite and less than or equal to $|\mathcal{S}|$.

The state-augmented MDP's state space $\mathcal{S}$ extends to the MDP with state-subgoal pairs $< \mathcal{S}, \mathcal{G} >$. Therefore, the policy based on the opponent's subgoal is updated as:

$$\mathcal{Q}(s_t, g_t, a_t) = \mathbb{E}_{\mathcal{P}(s_{t+1}|s_t,a^o,a)}[r + \gamma \max_a \mathcal{Q}(s_{t+1}, g_t, a)]. \tag{4}$$

Here $s_{t+1}, g_t$ is used instead of $s_{t+1}, g_{t+1}$, because of we assume that the next state of $s_t, g_t$ follows the same goal. In the framework of OMG, $g_t$ and $g_{t+1}$ will reach the same at the end of the episode.

---

**Algorithm 1** Opponent Modeling based on Subgoals Inference

---

1: **_Preparation:_**
2: Interact with $\nu$ opponents to collect $s$ and train the prior model $f_\psi$
3: Initialize subgoal inference model parameters $\phi$ and $\theta$
4: Initialize Q-network $\mathcal{Q}$ and the replay buffer $\mathcal{D}$
5: **repeat**
6:    **_Interaction phase_**
7:    Observe state $s$ and last opponent's action $a^o$
8:    Infer the subgoal $\hat{g}$ by subgoal inference model $q_\phi(g|\tau)$
9:    Choose action $a$ by $\max_a \mathcal{Q}(s, \hat{g}, a)$ with $\epsilon$-greedy
10:    Store trajectory experience $(s, a, a^o, r)$ in replay buffer $\mathcal{D}$
11:    **_Update phase_**
12:    **if** It's update time. **then**
13:       Calculate prior subgoal $\bar{g}$ by (6) or (7)
14:       Calculate subgoal $g$ by (8)
15:       Update Q-network by (4)
16:       Update subgoal inference model $q_\phi$ and $p_\theta$ by (5)
17:    **end if**
18: **until** convergence

---

To demonstrate the difference between learning Q-values using the opponent's action Equation (3) and using the opponent's subgoal Equation (4), we carry out an experiment in an $11 \times 11$ gridworld with two agents, as detailed in Figure 3. The Q-value using the opponent's action learns slower than the Q-value with the opponent's subgoal in Figure 3(a), resulting from the tuple $(s, a^o, a)$ is more numerous than $(s, g, a)$ in the Q-table. After convergence, the Q-value increases as it gets closer to the reward point, indicating a meaningful Q-value with the opponent's subgoal, as shown in Figure 3(b).

When there are fewer $(s, g, a)$ than $(s, a^o, a)$, the method using $(s, g, a)$ naturally holds the advantage of faster learning than the method of $(s, a^o, a)$. The quantity of $(s, g, a)$ is contingent upon the goal selection, and we present an analysis of the quantitative relationship between pair $(s, g)$ and $(s, a^o)$, see Appendix A.1. In short, the number of $(s, g)$ is significantly smaller than that of $(s, a^o)$ in our method.

## 3.3 Opponent Modeling based on Subgoal Inference

In this part, we elaborate on the opponent modeling module, which is divided into two components: _the subgoal inference model_ and _the subgoal selector_. The subgoal inference model utilizes the historical trajectory to predict opponent's subgoal, which act as the policy's input to make decisions during interaction phase. Meanwhile, the subgoal selector is responsible for scrutinizing the entire historical trajectory and choosing the suitable subgoal for training the subgoal inference model during the update phase.

**Subgoal inference model.** The subgoal $g$ is a representation of future states. Specifically, for a trajectory $\{s_0, a_0, a_0^o, \ldots, s_t, a_t, a_t^o, \ldots, s_T\}$. The state corresponding to subgoal $g_t$ is one of future states $\mathcal{N}_t = \{s_{t+1}, s_{t+2}, \ldots, s_T\}$, denoted as $s_t^g$. We denote the mapping between states and subgoals by $f_\psi$, where $\psi$ is the parameters and $\bar{g}_t = f_\psi(s_t^g)$.

The objective of the subgoal inference model is to infer $s_t^g$ from the historical trajectory $\tau_t = \{s_0, a_0, a_0^o, \ldots, s_{t-1}, a_{t-1}, a_{t-1}^o\}$ at timestep $t$, even though $s_t^g$ may be a state at timestep $t + 1$ or further. This is in accordance with the intuitive hypothesis, implying that the opponent's intention is often inferred after just a few initial actions.

Here, we introduce variational inference and use a conditional variational auto-encoder (CVAE) as the subgoal inference model. In this model, we represent the posterior probability as $q_\phi(\hat{g}|\tau)$ and the likelihood estimate as $p_\theta(\tau|g)$ with $\theta, \phi$ denoting network parameters. Additionally, the condition vector of the model is encoded using an RNN. The subgoal's prior model, denoted as $p_\psi(\bar{g}|s^g)$, is constructed using a pre-trained variational autoencoder (VAE), with the prior subgoal state $s^g$ being derived from the subgoal selector as its input. The distribution of subgoal prior $p_\psi$, subgoal

posterior probability $q_\phi$ and subgoal's prior $p_\psi(\bar{g}|s^g)$ are used normal distribution. The mapping $f_\psi$ represents sampling subgoal $\bar{g}$ from $p_\psi$ using reparameterization trick. The detailed network architecture is presented in Figure 2. The optimization objective of the subgoal inference model is:

$$< \hat{\theta}, \hat{\phi} >= \arg\max_{\theta,\phi} \mathbb{E}_{g \sim q_\phi(\hat{g}_t|\tau_t,s_t)} \Big[ \log p_\theta(s_t|\hat{g}_t,\tau_t) \Big] - \mathrm{KL}\Big( q_\phi(\hat{g}_t|\tau_t,s_t)||p_\psi(\bar{g}_t|s^g) \Big). \quad (5)$$

**Subgoal selector.** The objective of the subgoal selector is to choose the appropriate future state from $\mathcal{N}_t$ as prior model's input. The selection of subgoal states plays a pivotal role in shaping the behavior of an agent, as it significantly impacts the pattern of the agent's policy, either leaning towards optimism or conservatism. This critical decision-making process becomes especially pertinent when dealing with cooperative games and general-sum games, where the dynamics of interaction are complex and multifaceted. In these contexts, we provide two distinct manners to guide the agent's decision-making:

$$\bar{g}_t = \arg\max_{g \in f_\psi(\mathcal{N}_t^H)} V(s_t, g) \quad (6)$$

$$\bar{g}_t = \arg\min_{g \in f_\psi(\mathcal{N}_t^H)} V(s_t, g) \quad (7)$$

where $V(s,g) = \mathbb{E}_a Q(s,g,a)$, $\mathcal{N}_t^H$ is the set of future states $\{s_{t+1}, \cdots, s_{t+H}\}$. We use states within the next $H$ timesteps instead of all future steps because the subgoals of different trajectory fragments may have combinatorial properties. It gives the agent better generalization ability when facing different policy opponents. However, if we adopt the full horizon, the agent may prefer the goals near the terminal state, which is not conducive to the exploration of goal space.

When utilizing the subgoal $g$ as indicated in Equation (6), we pinpoint the state within a $H$-horizon that maximizes the V-value. The agent incorporates this as the subgoal to optimize the Q-function, thus adopting an optimistic strategy akin to the maximax strategy(Ben-Haim, 2006), which applies to cooperative games. Conversely, if we choose the subgoal as presented in Equation (7), it corresponds to the state yielding the lowest value. The agent then employs this as the subgoal for Q-function optimization, leading to a conservative strategy similar to the minimax strategy, which is usually used in general-sum games.

In conclusion, the subgoal selector and the subgoal inference model as a whole constitute the opponent modeling module. During the interaction phase, the subgoal inference model is used to get the inferred subgoal $\hat{g}$, which is combined with the state as the input to the Q-network. During the update phase, the prior subgoal $\bar{g}$ generated by the subgoal selector provides the inference model for training. When policy updating, the subgoal inference model is unstable at the beginning, which disturbs the updating of the Q-network. Therefore, we use the following combination of the prior subgoal $\bar{g}$ and the inferred subgoal $\hat{g}$,

$$g_t = \hat{g}_t \mathbb{I}(\eta > \epsilon) + \bar{g}_t \mathbb{I}(\eta \le \epsilon), \quad \eta \sim U[0,1], \quad (8)$$

where $\epsilon$ is a hyperparameter that decreases to zero over training.

For completeness, the full procedure of OMG is given in Algorithm 1.

## 4 EXPERIMENTS

First, we evaluate OMG's training performance in two environments (discrete and continuous state spaces) and then test its generalization against opponents with various policies in a complex environment. In all the experiments, the baselines have the same neural network architectures as OMG. All the methods are trained for five runs with different random seeds, and results are presented using mean and standard deviation. More details about experimental settings and hyperparameters are available in Appendix A.2.

### 4.1 MULTI-AGENT ENVIRONMENTS

**Foraging** environment (Albrecht & Ramamoorthy, 2015; Albrecht & Stone, 2019) is an $8 \times 8$ grid-world containing two players: the agent and the opponent. At the beginning of each round, the

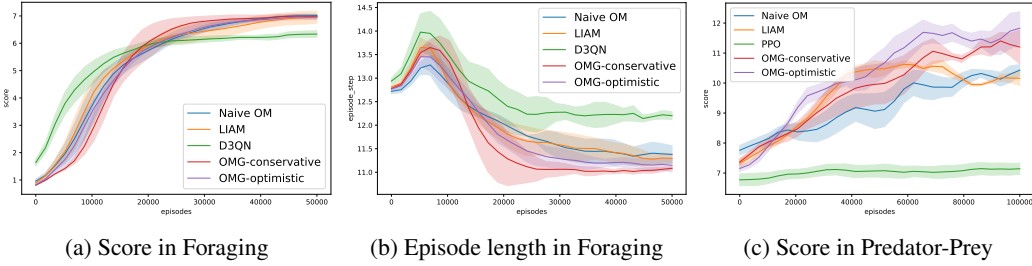

(a) Score in Foraging      (b) Episode length in Foraging      (c) Score in Predator-Prey

Figure 4: Training performance in Foraging and Predator-Prey. (a) shows the total score obtained by the agent. (b) illustrates the number of steps at the end of each episode. The results show that OMG can converge to the same score as baselines but end the episode in fewer steps because it predicts the opponent's goal. (c) shows the score obtained by the agent as a predator with two other uncontrolled predators in Predator-Prey, and OMG outperforms the baselines.

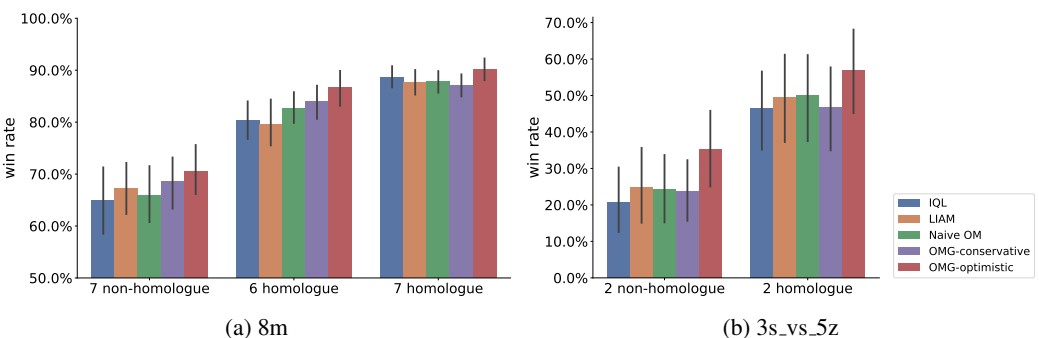

(a) 8m                 (b) 3s_vs_5z

Figure 5: Test performance of cooperation with different opponents in *8m* and *3s_vs_5z* maps of SMAC. The results show that OMG-optimistic outperforms all baselines. The results are averaged over collaborating with 30 opponents of different policies, with 95% confidence intervals.

players and three foods are randomly generated in the environment. The goal of the agent is to collect all foods as quickly as possible. The agent can move in four directions or pick up the food. The agent must judge the opponent's target food as soon as possible to avoid futile actions for the same food.

**Predator-Prey** (Lowe et al., 2017) is a three-against-one multi-agent environment with a continuous space. Three predators coordinate to touch the prey. The agent acts as one of the predators, and the opponents are the other two predators and the prey, which leads to the non-stationarity of the environment from the agent's view despite not belonging to one camp. The agent aims to maximize its reward and therefore needs to collaborate with the other two predators to complete the encirclement and cut the prey's escape route.

**SMAC** (Samvelyan et al., 2019) is a high-dimensional complex environment for research in the field of collaborative MARL based on StarCraft II. The agent joins a set of agents with unknown policies to accomplish the task. The only way to accomplish the task is to collaborate with the other agents. The agent's goal is to complete the task with a group of opponents controlled by unknown policies.

## 4.2 BASELINES

In the experiments, we implement two variants of OMG, OMG-optimistic and OMG-conservative, based on the subgoal selection patterns in Equation (6) and Equation (7), respectively. OMG compared with the following methods:

- Naïve OM (He et al., 2016) uses observation to directly model the opponent's policy, which assists the agent in decision-making by predicting the opponent's actions.

- LIAM (Papoudakis et al., 2021) uses the observations and actions of the modeling agent with an encoder-decoder architecture, and the model learns to extract representations about the modeling agent, conditioned only on the local observations of the controlled agent.
- D3QN & PPO & IQL (Wang et al., 2016; Schulman et al., 2017; Tampuu et al., 2017) are classical RL algorithms without opponent modeling.

We use D3QN, PPO, and IQL as the backbone algorithms in Foraging, Predator-prey, and SMAC, respectively, to reproduce the performance of baselines. The versions of OMG that are based on D3QN and IQL incorporate "dueling" and "double" tricks over Algorithm 1. For OMG based on PPO, please refer to Appendix A.3 for details.

## 4.3 PERFORMANCE OF TRAINING

We evaluate the performance of OMG on foraging and predator-prey, and the results are shown in Figure 4. In the foraging environment, our method attains comparable scores to the baseline methods, and both the agent and the opponent achieve similar scores. OMG has a shorter episode length compared to other methods as demonstrated in Figure 4(b), because OMG can predict the subgoal that the opponent is heading to and thus avoid wasting steps in the same direction. In addition, the results show that OMG-conservative is more suitable than OMG-optimistic in this scenario since this is a general-sum game. The action modeling-based methods, LIAM and Naïve OM, demonstrate comparable performance, whereas D3QN without opponent modeling, exhibits subpar results. In the predator-prey environment, the agent acts as the predator and collaborates with the other two uncontrolled predators to catch the prey. The results in Figure 4(c) show that OMG obviously learns faster than action modeling methods, which demonstrates that OMG can also work efficiently in continuous state space. PPO without opponent modeling can hardly improve performance in training due to the non-stationarity caused by opponents. OMG-optimistic performs better than OMG-conservative because OMG-optimistic is suitable for the cooperative game.

## 4.4 GENERALIZATION TO UNKNOWN OPPONENTS

We evaluate the generalization of OMG in a complex multi-agent environment, SMAC, which enables the opponent to exhibit more diverse policies. The experimental results of *8m* and *3s_vs_5z* are shown in Figure 5. The test set consists of 30 opponents with different policies, trained by the IQL, VDN(Sunehag et al., 2017), and QMIX(Rashid et al., 2020). In *8m*, the opponents are reorganized into three groups: *7 homologues*, *6 homologues*, and *7 non-homologues*. In *3s_vs_5z*, the opponents falls into two groups: *2 homologues* and *2 non-homologues*. Here, *homologue* refers to the policy from the same algorithm with the same parameters, and *non-homologue* represents the policy from two different algorithms. The remained agents are controlled by OMG or baseline algorithms. Without opponent modeling, IQL struggles to adapt to various opponents, resulting in poor performance, especially when the opponent is *non-homologue*. This underscores the effectiveness of opponent modeling in autonomous agent tasks. LIAM and Naïve OM are the opponent's action modeling methods that contributed to the team's improved win rate to some extent. The mediocre performance of OMG-conservative is attributed to its overly cautious subgoal selection, but there is no significant performance drop compared to IQL, which is consistent with the "conservative". OMG-optimistic surpasses the baseline methods in cooperative tasks. OMG-optimistic cooperates with unknown opponents through positive subgoal selection, which is easier to win in hard scenarios. For opponents and training details, please refer to Appendix A.2.

## 4.5 ABLATION STUDY

The results of the ablation study in Foraging are presented in Figure 6. Specifically, Figure 6(a) and Figure 6(b) correspond to experiments related to subgoal selection. During the policy update, Equation (8) (i.e., $g$) is utilized. As $f_\psi$ is pre-trained and fixed during the update phase, $\bar{g}$ remains stable. On the other hand, $\hat{g}$, which represents the inferred subgoal when executing the policy, also stabilizes as the training steps increase. The transition of $g$ from $\bar{g}$ to $\hat{g}$ is a gradual process, which helps avoid instability during the training of the subgoal inference model.

The parameter $H$ denotes the horizon of the subgoal selector. The ablation experiment results are shown in Figure 6(c) and Figure 6(d). It is observed that an appropriate horizon value is neither

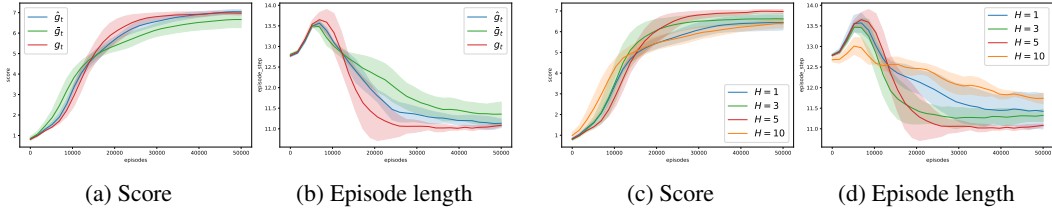

| (a) Score | (b) Episode length | (c) Score | (d) Episode length |

Figure 6: Ablation study of OMG in Foraging. (a) and (b) compares OMGs with different subgoal learning policy. (c) and (d) show ablation study for the hyperparameter horizon $H$.

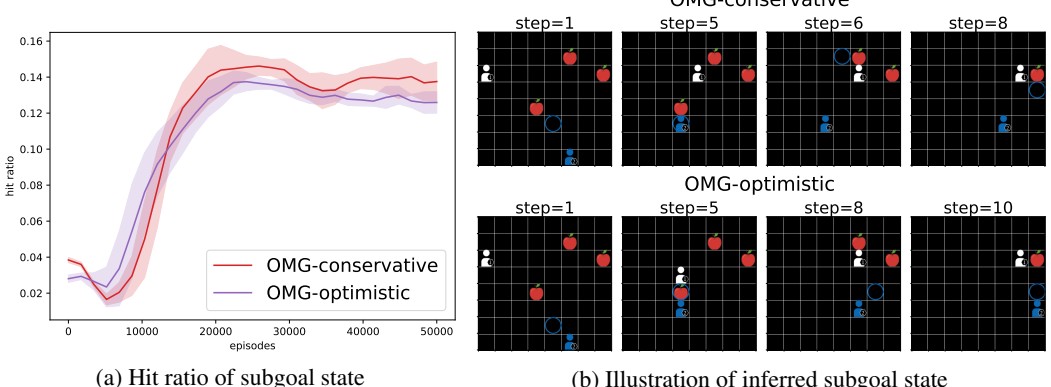

| (a) Hit ratio of subgoal state | (b) Illustration of inferred subgoal state |

Figure 7: Subgoal analysis of OMG in Foraging. The subgoal hit rates for OMG-conservative and OMG-optimistic are shown in Figure 7(a). In Figure 7(b), a blue circle represents the state obtained through the reconstruction of the subgoal inferred by the agent. The figure illustrates the difference between OMG-conservative and OMG-optimistic under the same initial state and opponent policy.

excessively high nor excessively low. When $H = 1$, it is essentially equivalent to combining with QSS (Edwards et al., 2020) and opponent modeling. However, if $H$ is set too high, such as $H = 10$, the agent may skip important states in the trajectory, leading to a degradation in performance. Therefore, selecting an appropriate value for $H$ is crucial in achieving satisfactory results.

## 4.6 INFERRED SUBGOAL ANALYSIS

In Figure 7(a), we plot the ratio of that an opponent's future trajectory passes through the opponent's subgoal inferred by the agent, termed subgoal hit ratio. The subgoal hit ratio is calculated by reconstructing the subgoal state $f_\psi^{-1}(\hat{g})$. The subgoal hit rate gradually improves during training, which indicates that the subgoal-based opponent modeling is able to predict the future state of the opponent. OMG tends to predict goals multiple steps ahead, making it difficult for opponents to reach immediately, resulting in a modest value that hit ratio convergence. There is a small gap between the subgoal hit rates of OMG-conservative and OMG-optimistic, which leads to longer episode length for OMG-optimistic than OMG-conservative, as illustrated in Figure 7(b). The root cause lies in the differences in subgoal selection between OMG-conservative and OMG-optimistic.

## 5 CONCLUSION

In this work, we introduce OMG, a novel method for opponent modeling based on subgoal inference. OMG is a simple and efficient opponent modeling method and can be combined with different RL algorithms. Unlike most opponent modeling methods, which primarily focus on predicting the opponent's actions, OMG focuses on modeling the opponent's subgoals. Specifically, it leverages the value function of the policy to guide the selection of subgoals, which yields two variants of OMG for cooperative and general-sum games, respectively. Empirical results demonstrate the remarkable performance achieved by OMG, as compared to baselines which are based on action modeling, and

that OMG exhibits better generalization when cooperating with opponents with unknown policies. We analyze the subgoals obtained by the inference model, and the results show that they closely correlate with the opponent's trajectory. The limitation of OMG is it cannot handle open multi-agent systems where agents may enter and leave the system during the interaction. This is left for our future work.

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

# A APPENDIX

## A.1 ANALYSIS OF $(s, g)$

In opponent modeling, we usually build $(s, g)$ and $(s, a^o)$ by observing the opponent's action trajectories. We construct a tree to describe the trajectories of the opponent's action sequences, as Figure 8. The non-leaf nodes and edges represent the state and opponent's action respectively. Without loss of generality, we simplify the problem by using a complete tree with the leaf node as goal. The length of the action sequences is $k$ and the opponent action space is denoted as $A$. We compare the number of $(s, a)$ and $(s, g)$ that can be observed via trajectories, and their sets are denoted as $\mathcal{S}_a$ and $\mathcal{S}_g$ respectively. The sizes of $\mathcal{S}_a$ and $\mathcal{S}_g$ as:

$$card(\mathcal{S}_a) = \sum_{l=0}^{k-1} \sum_{s \in \mathcal{S}^{(l)}} n_A = \frac{n_A^k - 1}{n_A - 1} n_A$$

$$card(\mathcal{S}_g) = \sum_{l=0}^{k-1} \sum_{s \in \mathcal{S}^{(l)}} \sum_{g \in G} \mathbb{I}(s \to g)$$

$$\leq |G| + n_A \cdot \frac{|G|}{n_A} + \cdots + n_A^{k-1} \cdot \frac{|G|}{n_A^{k-1}}$$

$$= k|G|$$

where $\mathcal{S}^{(l)}$ represents the set of all states of depth $l$ in the tree. $s \to g$ means $g$ is reachable from $s$. $n_A$ is the size of $A$. Let $card(\mathcal{S}_g) \leq card(\mathcal{S}_a)$, we get a bound over $|G|$, as (9). When the goal number of our method is within the bound, the number of expanded states can be significantly reduced, which means the RL algorithm learns faster than those methods based on action modeling.

$$card(\mathcal{S}_g) \leq card(\mathcal{S}_a) \Rightarrow |G| \leq \frac{n_A}{k} \frac{n_A^k - 1}{(n_A - 1)} = \frac{n_A}{k} |\mathcal{S}| \tag{9}$$

When $|G|$ is below $n_A/k$ times the number of observed states, the goal-based opponent modeling method proves more advantageous compared to the methods based on action modeling. Consequently, this criterion can be met by maintaining a relatively modest value for $k$. Due to our method favoring the adoption of extreme values as goal states, a limited quantity of such states exist. So, it is loosely bound of $|G|$ for OMG.

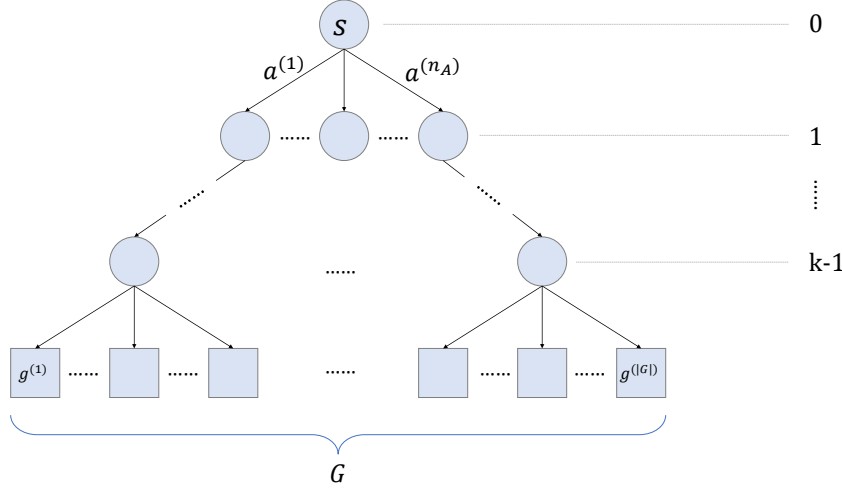

Figure 8: Illustration of opponent's decision tree. Circles, edges, and squares represent state nodes, action, and goal nodes respectively.

## A.2 EXPERIMENTS SETTINGS

**Opponent.** The autonomous agent is trained in a multi-agent environment, where it interacts with the opponents controlled by a set of pre-trained policies. At the onset of each episode, the opponent's policy is selected randomly from the set. In the case of SMAC, the autonomous agent's index is also randomly determined. For Foraging, Predator-Prey, and SMAC environments, D3QN, PPO, and QMIX are used to train the opponents, respectively. All the opponents in the training set comprise 10 distinct policies.

When assessing the performance of the autonomous agent in the SMAC with a test set, these opponents in the set are trained separately using IQL, VDN, and QMIX, with 10 instances for each training method. To illustrate the dissimilarity of the test opponent's policies, we utilize a set of identical states to acquire the action vectors of the policy in the test set. We visualize the action vectors, as demonstrated in Figure 9. The figure shows the diversity of test set policies employed by the test opponents. The test results are averaged over 100 episodes of fine-tuning, with 5 random seeds.

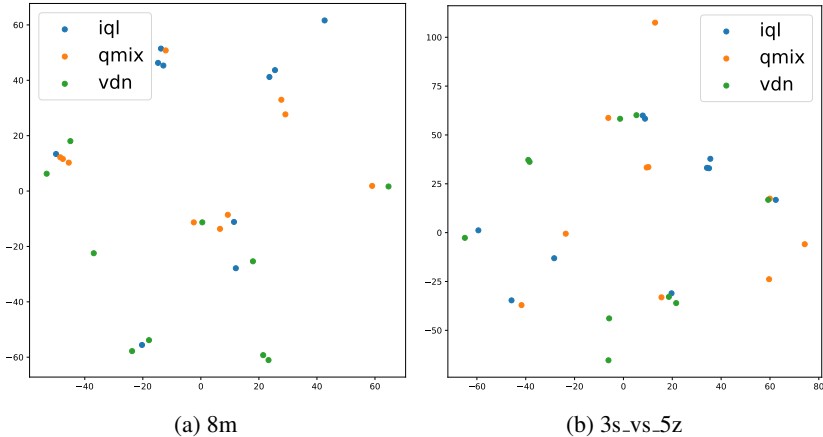

(a) 8m             (b) 3s_vs_5z

Figure 9: The distribution of the opponent's policy for the test of generalization.

**Pre-train the subgoal's prior model.** The subgoal's prior model $p_\psi(\bar{g}|s^g)$ is a VAE that learns from a set of states that are collected while training opponents. The optimization objective of VAE is :

$$<\hat{\omega}, \hat{\psi}> = \underset{\omega,\psi}{\arg\max} \, \mathbb{E}_{g \sim q_\psi(g|s)} \Big[ \log p_\omega(s|g) \Big] - \mathrm{KL}\Big( q_\psi(g|s) || \mathcal{N}(0,1) \Big). \tag{10}$$

where $\psi$ and $\omega$ are parameters of the encoder and the decoder, respectively. The decoder $p_\omega(s|g)$, also denoted by $f_\psi^{-1}$, is also used to reconstruct the subgoal state, as shown in Section 4.6.

**Hyperparameters.** All hyperparameters are listed in the table below:

Table 1: Hyperparameters

| | | Q-based RL | Foraging(D3QN) | SMAC(IQL) | Policy-based RL | Predator-prey(PPO) |
|---|---|---|---|---|---|---|
| RL Algorithm | hidden units | | MLP[64, 32] | RNN[64, 64] | hidden units | MLP[64, 32] |
| | activation function | | ReLU | ReLU | activation function | ReLU |
| | optimizer | | Adam | RMSProp | optimizer | Adam |
| | learning rate | | 0.005 | 0.0005 | learning rate | 0.0005 |
| | target update interval | | 100 | 200 | num. of updates | 10 |
| | epsilon start | | 0.5 | 0 | value discount factor | 0.99 |
| | epsilon end | | 0.95 | 0.95 | GAE parameter | 0.99 |
| | epsilon anneal time | | 4500 | 50000 | clip parameter | 0.115 |
| | batch size | | 32 | 32 | max grad norm | 0.5 |
| | buffer size | | 5000 | 5000 | | |
| Opponent model | hidden units | | MLP[64, 32] | MLP[64, 32] | | MLP[64, 32] |
| | learning rate | | 0.001 | 0.001 | | 0.001 |
| | subgoal horizon | | 5 | 10 | | 5 |
| | KL weight | | 0.001 | 0.001 | | 0.001 |
| | $\Delta \eta$ | | 0.001 | 0.001 | | 0.001 |
| | $\epsilon$ start | | 0.5 | 0.5 | | 0.5 |
| | $\epsilon$ anneal time | | 50000 | 50000 | | 50000 |

## A.3 OMG BASED ON PPO

---

**Algorithm 2** OMG based on PPO

---

1: *Preparation:*
2: Interact with $\nu$ opponents to collect $s$ and train the prior model $f_\psi$
3: Initialize subgoal inference model parameters $\tau$ and $\theta$
4: Initialize policy parameters $\delta$ and value function parameters $\varphi$
5: **for** k=0,1,2,... **do**
6:    *Interaction phase*
7:    Observe state $s$ and last opponent's action $a^o$
8:    Infer the subgoal $\hat{g}$ by subgoal inference model $q_\phi(g|\tau)$
9:    Choose action $a$ by $\pi_{\delta_k}(\cdot|s, \hat{g})$
10:    Store experience $(s, a, a^o, r)$ in buffer $\mathcal{D}_k$
11:    *Update phase*
12:    Calculate prior subgoal $\bar{g}$ by Equation (6) or Equation (7)
13:    Calculate subgoal $g$ by Equation (8)
14:    Update policy parameters by

$$\delta_{k+1} = \arg\max_\delta \frac{1}{|\mathcal{D}_k|T} \sum_{\tau \in \mathcal{D}_k} \sum_{t=0}^{T} \min\left(\frac{\pi_\delta(a_t|s_t)}{\pi_{\delta_k}(a_t|s_t)} A^{\pi_{\delta_k}}(s_t, a_t), g(\epsilon, A^{\pi_{\delta_k}}(s_t, a_t))\right) \quad (11)$$

15:    Update value parameters by

$$\varphi_{k+1} = \arg\min_\varphi \frac{1}{|\mathcal{D}_k|T} \sum_{\tau \in \mathcal{D}_k} \sum_{t=0}^{T} (V_\varphi(s_t) - \hat{R}_t)^2 \quad (12)$$

16:    Update inference model $q_\phi$ and $p_\theta$ by Equation (5)
17: **end for**

---

