# OpenReview forum: "Opponent Modeling based on Sub-Goal Inference"
_ICLR.cc/2024/Conference — Submitted to ICLR 2024_

### Official Review · Reviewer_8YJ7 · 2023-10-23

**Soundness:** 2 fair
**Presentation:** 2 fair
**Contribution:** 2 fair
**Rating:** 3
**Confidence:** 4

**Summary:**

This paper focuses on the opponent modeling problem and proposes an opponent modeling algorithm based on sub-goal inference. Concretely, the proposed method designs a subgoal inference model and a subgoal selector. The subgoal inference model takes historical trajectory as input and outputs the predicted subgoal (a high-dimensional vector) of the opponents. It is trained based on variational inference. The subgoal selector chooses a subgoal for the update of the Q-value network. It aids the training of the subgoal inference model as the Q-value network affects the sampling process.

**Strengths:**

+ This paper introduces the subgoal concept from MARL to the opponent modeling domain, which is promising.

+ The experiments are performed in three experiments (although two of them are very simple scenarios).

**Weaknesses:**

- The paper lacks clarity and there are some confusing parts that need more explanation.

- Some technical designs of the proposed method do not conform with the canonical RL algorithm. More theoretical analysis is needed.

- As shown in Figure 7(a), the prediction accuracy of opponent subgoal is low.

**Questions:**

1. In Section 3.2, this paper adopts a state-extended MDP. Therefore, for the next state, the Q-value equation should be $Q(s_{t+1},g_{t+1}, a)$. However, the Eq. (4) writes it as $Q(s_{t+1},g_{t}, a)$. It should be explained that why $g_t$ is the same as $g_{t+1}$.

2. In line 3 of Algorithm 1, the subgoal inference model parameters should be $\phi$ and $\theta$ instead of $\tau$ and $\theta$.

3. In algorithm 1, the action is sampled based on $Q(s,\hat{g},a)$ given $\hat{g}$ (line 9). However, when updating Q-network, the subgoal may be $\bar{g}$ instead of $\hat{g}$ based on Eq. (8). That means given the same experience tuple $(s, a, a^o, r)$, the Q-network input may be different for the sampling and learning phase, which does not conform with the canonical RL theory. What would the theoretical impacts be?

4. In Eq. (5), $\bar{g}$ is output by a prior model $p_{\psi}$ while in Eq. (6) and Eq. (7), it is from $f_{\psi}$. What is the relationship between $p_{\psi}$ and $f_{\psi}$? Moreover, when $f_{\psi}$ first appears in Page 5, it outputs $\bar{g}_t$ given $s_t^g$. However, in Eq. (6) and Eq. (7), its input format becomes $\mathcal{N}_t^H$. Why?

5. Why in a general-sum game,  a conservative strategy is more suitable? Given the opponents' subgoal, shouldn't we just maximize our own Q-value?

6. How to get the $f^{-1}_{\psi}$ in Section 4.6?

7. How to compute $card(\mathcal{S}_g)$ in Appendix A.1? Based on my understanding, if we assume every state can be transit to a goal state after a certain action sequence, $card(\mathcal{S}_g)$ should be $\frac{n_A^k-1}{n_A-1}|G|$. In addition, the sentence "Moreover, a lower $k$ value, signifying predicted subgoals, facilitates generalization. So, it is loosely bound of $|G|$ for OMG. Due to our method favoring the adoption of extreme values as goal states, a limited quantity of such states exist" is confusing.

8. In appendix A. 2, this paper uses a set of identical states to acquire the action vectors of the policy in the test set. What is the detailed process of obtaining the set of identical states? Are these states sampled by a certain policy?

9. The reason of baseline method choice is unclear. Why not compare with some newer opponent modeling methods, e.g., [1] and [2]?

[1] Greedy when sure and conservative when uncertain about the opponents. ICML 2022.

[2] Model-based opponent modeling. NeurIPS 2022.

---

> ### Author Response · Authors · 2023-11-23
> **Response to Reviewer 8YJ7 （Part Ⅰ）**
>
> Thanks for your comments. The minor issues you raised, we have addressed them in the revision. As follows, we address your concerns in detail.
>
> > In Section 3.2, this paper adopts a state-extended MDP. Therefore, for the next state, the Q-value equation should be $Q(s_{t+1}, g_{t+1}, a)$. However, the Eq. (4) writes it as $Q(s_{t+1}, g_{t}, a)$. It should be explained that why $g_t$ is the same as $g_{t+1}$.
>
> We extend the state space of the MDP with state-subgoal pairs to form a state-augmented MDP $\mathcal{S}$. We assume that the next state of $s_t, g_t$ follows the same goal, so it is $s_{t+1}, g_t$. The $g_t$ is selected from $\mathcal{N}_t$. When the trajectory terminates, $g_t$ and $g_{t+1}$ reach the final state.
>
> > In algorithm 1, the action is sampled based on $Q(s,\hat{g},a)$ given $\hat{g}$ (line 9). However, when updating Q-network, the subgoal may be $\bar{g}$ instead of $\hat{g}$ based on Eq. (8). That means given the same experience tuple $(s, a, a^o, r)$, the Q-network input may be different for the sampling and learning phase, which does not conform with the canonical RL theory. What would the theoretical impacts be?
>
> The hyperparameter $\epsilon$ is decreased during training. The probability of $g$ taking $\hat{g}$​ increases linearly up to 50,000 episodes. It does produce better results than just using $\hat{g}$​, as shown in Section 4.5.
>
> We do not focus on the theoretical analysis of this problem, because the non-stationary problem is complex and the inference error of opponent modeling is intractable and unavoidable.
>
> > In Eq. (5), $\bar{g}$ is output by a prior model $p_{\psi}$ while in Eq. (6) and Eq. (7), it is from $f_{\psi}$. What is the relationship between $p_{\psi}$ and $f_{\psi}$? Moreover, when $f_{\psi}$ first appears in Page 5, it outputs $\bar{g}_t$ given $s_t^g$. However, in Eq. (6) and Eq. (7), its input format becomes $\mathcal{N}_t^H$. Why?
>
> In essence, $f_\psi$ and $p_\psi$ are the same functions. The subgoal probability uses a normal distribution, with $p_{\psi}(\bar{g}|s^g)$ describes the probability distribution of subgoals. The mapping from state $s^g$ to subgoal $\bar{g}$ is expressed as $\bar{g} = f_\psi(s^g)$. This process involves encoding the state $s^g$ into the distribution and sampling from the distribution to derive $\bar{g}$. Apologies for the confusion, we makes it clearer in the revision.
>
> > Why in a general-sum game, a conservative strategy is more suitable? Given the opponents' subgoal, shouldn't we just maximize our own Q-value?
>
> There are some misunderstandings here. That is the intention of the opponent, that to maximize or minimize our Q-value, which is just used to choose opponents' subgoal. The agent's policy is always to maximize the Q-value.
>
> > How to get the $f^{-1}_{\psi}$ in Section 4.6?
>
> $f^{-1}_{\psi}$refers to the decoder in the VAE, which is trained using the states data in the pre-training phase.
>
> > How to compute $card(\mathcal{S}_g)$ in Appendix A.1? Based on my understanding, if we assume every state can be transit to a goal state after a certain action sequence, $card(\mathcal{S}_g)$ should be $\frac{n_A^k-1}{n_A-1}|G|$. In addition, the sentence "Moreover, a lower $k$ value, signifying predicted subgoals, facilitates generalization. So, it is loosely bound of $|G|$ for OMG. Due to our method favoring the adoption of extreme values as goal states, a limited quantity of such states exist" is confusing.
>
> It's important to note that not every state can be transit to **every** goal, so $\frac{n_A^k-1}{n_A-1}|G|$ is not correct. Intuitive explanation for $card(\mathcal{S}_g) \leq k|G|$ is that each goal as a leaf node in a tree of depth k, have at most k ancestor state. Only each ancestor state can transition to the corresponding goal state after a certain sequence of actions. Therefore, there are at most $k|G|$ of these ancestor states.
>
> We modified the confusing statement in the revision, as follows: "When $|G|$ is below $n_A/k$ times the number of observed states, the goal-based opponent modeling method proves more advantageous compared to the methods based on action modeling. Consequently, this criterion can be met by maintaining a relatively modest value for $k$. Due to our method favoring the adoption of extreme values as goal states, a limited quantity of such states exist. So, it is loosely bound of $|G|$ for OMG."

---

> ### Author Response · Authors · 2023-11-23
> **Response to Reviewer 8YJ7（Part Ⅱ）**
>
> > In appendix A. 2, this paper uses a set of identical states to acquire the action vectors of the policy in the test set. What is the detailed process of obtaining the set of identical states? Are these states sampled by a certain policy?
>
> These states are stored when the policies in the test set are used, just to verify that the policies in the test set are the same. The states can be collected with any policy, and this does not affect the results.
>
> > The reason of baseline method choice is unclear. Why not compare with some newer opponent modeling methods, e.g., [1] and [2]?
> >
> > [1] Greedy when sure and conservative when uncertain about the opponents. ICML 2022.
> >
> > [2] Model-based opponent modeling. NeurIPS 2022.
>
> OMG is a new exploration in the opponent modeling method, which is different from the previous opponent modeling ideas, so we choose the classic opponent modeling methods as the baselines.
>
> [1] was not suitable for our testing environment for two reasons. Firstly, it is designed to work in a competitive environment. Secondly, it is essentially a fusion of LIAM-VAE and EXP3, where EXP3 is a bandit algorithm.
>
> [2] is a method to deal with a constantly learning opponent. When the opponent’s policy remains unchanged, the method that degenerates to naive action of opponent modeling yields similar results to Naive OM in baselines.

---

### Official Review · Reviewer_W6q1 · 2023-10-31

**Soundness:** 2 fair
**Presentation:** 3 good
**Contribution:** 1 poor
**Rating:** 3
**Confidence:** 5

**Summary:**

This paper proposes an opponent modeling method (OMG) based on subgoal inference. In particular,  a subgoal inference model is employed to infer the subgoal of the opponent based on a VAE architecture. A subgoal selector model is used to make a balance between the inferred subgoal and a prior subgoal. In addtion, the agent policy or value function is conditioned additionally on the selected subgoal. Experimental studies on standard benchmarks verify the effectiveness of the proposed method.

**Strengths:**

- The paper is clearly presented.
- The experimental study seems comprehensive.

**Weaknesses:**

Main weaknesses:

- I don't think predicting subgoals itself alone can be claimed as a significant contribution in comparison to previous work on opponent modelling.  [1] predicts the opponent’s goal as well. Also, the subgoals discussed in OMG are essentially future states of the opponent, and predicting future states are not necessarily better than predicting future actions.

- I don't think it is a correct statement to say that most previous work on opponent modeling predicting the opponent's actions. For instance, [2] predicts a policy embedding. [3] is a meta-learning method for opponent modelling.  [4] predicts which type of policy.

- The general architecture of OMG that uses VAE to predict something is quite similar to many previous opponent modelling methods that use VAE as well: for instance [5] and [6]


Minor:

- very related work on opponent modeling are missing in the literature review. For instance, [2] [3] [4]. A full literature review on opponent modeling is suggested.
- section 3.1 is too preliminary to be included in the Method section.
- the last sentence of section 3.2. "In short, the number of (s,g) is significantly smaller than that of (s,a)". Please provide strong evidence for this, as I do think it is domain-dependent whether the number of (s,g) is larger or smaller than (s,a).
- Equation 8 is very ad hoc.

[1] Roberta Raileanu, Emily Denton, Arthur Szlam, and Rob Fergus. Modeling others using oneself in
multi-agent reinforcement learning. In International conference on machine learning, pp. 4257–
4266. PMLR, 2018.

[2] Haobo Fu, Ye Tian, Hongxiang Yu, Weiming Liu, Shuang Wu, Jiechao Xiong, Ying Wen, Kai Li, Junliang
Xing, Qiang Fu, et al. Greedy when sure and conservative when uncertain about the opponents. In
International Conference on Machine Learning, pages 6829–6848. PMLR, 2022.

[3] Al-Shedivat, M., Bansal, T., Burda, Y., Sutskever, I., Mordatch, I., and Abbeel, P. Continuous adaptation via metalearning in nonstationary and competitive environments. In International Conference on Learning Representations, 2018

[4] Zheng, Y., Meng, Z., Hao, J., Zhang, Z., Yang, T., and Fan, C. A deep bayesian policy reuse approach against non-stationary agents. In Proceedings of the 32nd International Conference on Neural Information Processing
Systems, pp. 962–972, 2018.

[5] Georgios Papoudakis and Stefano V Albrecht. Variational Autoencoders for Opponent Modeling in
Multi-Agent Systems. arXiv preprint arXiv:2001.10829, 2020.

[6] Luisa Zintgraf, Sam Devlin, Kamil Ciosek, Shimon Whiteson, and Katja Hofmann. Deep interactive
bayesian reinforcement learning via meta-learning. arXiv preprint arXiv:2101.03864, 2021.

**Questions:**

Except for the experimental results, I am not convinced why predicting subgoals (i.e., future states) as OMG does are better than predicting something else about the opponents (e.g., actions, policy embeddings, policy categories, etc.). Could you please provide more justifications on this?

---

> ### Author Response · Authors · 2023-11-23
> **Response to Reviewer W6q1**
>
> Thanks for your comments. Below is our detailed explanation of your concerns.
>
> > I don't think predicting subgoals itself alone can be claimed as a significant contribution in comparison to previous work on opponent modelling. [1] predicts the opponent’s goal as well. Also, the subgoals discussed in OMG are essentially future states of the opponent, and predicting future states are not necessarily better than predicting future actions.
>
> The immediate idea is that predicting state contains more information than predicting action. We must acknowledge the fact that each method has its specific conditions of application and also has certain limitations. There is no guarantee that OMG will perform better than any method predicting future actions in any environment. In this paper, we empirically demonstrate that OMG has a general advantage over methods that predict actions of opponents. In addition, the method of [1] is to model the opponent's goal, but these goals need to be manually preset, discrete and low-dimensional, which obviously cannot be applied in such environments as Predator-Prey and SMAC.
>
> Intuitively, predicting future states of the opponent have an advantage over predicting future actions. Just like our example in Sec 1 Para 3:
>
> *For example, to reach the goal point of $(2, 2)$, an opponent moves from $(0, 0)$ following the action sequence $<\uparrow,\uparrow,\rightarrow,\rightarrow>$ by four steps (Cartesian coordinates). There are also 5 other action sequences, \textit{i.e.,} $<\uparrow,\rightarrow,\uparrow,\rightarrow>, <\uparrow,\rightarrow,\rightarrow,\uparrow>, <\rightarrow,\uparrow,\uparrow,\rightarrow>, <\rightarrow,\uparrow,\rightarrow,\uparrow>, <\rightarrow,\rightarrow,\uparrow,\uparrow>$, that can lead to the same goal. Obviously, the complexity of the action sequence is much higher than the goal itself.*
>
> Similar conclusions are also verified by the experiments in Figure 3, and the corresponding analysis is given in Appendix A.1.
>
> > I don't think it is a correct statement to say that most previous work on opponent modeling predicting the opponent's actions. For instance, [2] predicts a policy embedding. [3] is a meta-learning method for opponent modelling. [4] predicts which type of policy.
> > very related work on opponent modeling are missing in the literature review. For instance, [2] [3] [4]. A full literature review on opponent modeling is suggested.
>
> We revise this statement to be more rigorous and add those related work in the revision.
>
> > The general architecture of OMG that uses VAE to predict something is quite similar to many previous opponent modelling methods that use VAE as well: for instance [5] and [6].
>
> From the level of method and algorithm idea, our method OMG has no similarity with [5] [6]. It is just that the same VAE network is used, but for the different purpose.
>
> Firstly, [6] is the paradigm of CTDE, which is different from our setting of Autonomous agent. Although VAE is used in [6], it is still an action modeling method in essence. [5] is LIAM's previous work, which used supervised learning methods to construct mapping $<o_t, a_{t-1}, r_{t-1}, d_{t-1}>$ to $<a_t^{-1}>$ via VAE. OMG use VAE to encode future state as subgoal prior. It is worth noting that the future state is associated with the Q-value, which is novelty.
>
> > the last sentence of section 3.2. "In short, the number of (s,g) is significantly smaller than that of (s,a)". Please provide strong evidence for this, as I do think it is domain-dependent whether the number of (s,g) is larger or smaller than (s,a).
>
> "In short, the number of (s,g) is significantly smaller than that of (s,a)" does not always hold, depending on the method of GOAL selection. In the framework of our method, this is a reasonable conclusion. We prove our point in Appendix A.1. It is a well-known fact that no one method is suitable for all domains. Although OMG is not a panacea, its conditions are not strict and verified by experiments.
>
> > Except for the experimental results, I am not convinced why predicting subgoals (i.e., future states) as OMG does are better than predicting something else about the opponents (e.g., actions, policy embeddings, policy categories, etc.). Could you please provide more justifications on this?
>
> Your examples of opponent modeling methods [1]-[6] have all been verified by experimental results to demonstrate the effectiveness of the algorithm, which is the standard paradigm in the field of opponent modeling. We conducted comprehensive experimental verification by comparing with modeling action of opponent methods (Sec 3.2), performance of training (Sec 4.3), generalization to unknown opponents (Sec 4.4), and inferred subgoal analysis(Sec 4.6). The conclusions also show that our method has certain advantages than modeling action of opponent methods. In addition, our method is intuitive, as agreed by other reviewer.

---

### Official Review · Reviewer_ZFnJ · 2023-11-01

**Soundness:** 3 good
**Presentation:** 2 fair
**Contribution:** 3 good
**Rating:** 6
**Confidence:** 3

**Summary:**

This paper proposes an approach to opponent modelling based on predicting the opponent's subgoal rather than its next granular action.  (In this work, "subgoal" just means a desired future state.) The authors argue that this approach is simpler and more efficient than predicting the next action, since it abstracts away unimportant details and yields longer lasting predictions. A subgoal predictor is trained based on past trajectories using variational inference. The training label (i.e., the "true" subgoal that the opponent was previously following) is derived via maximax/minimax-style logic, depending on whether the scenario is cooperative/competitive. Experiments across multiple domains show that this yields a persistent edge versus previous approaches.

**Strengths:**

Assuming that I actually understood the paper, I think it contains some very clever ideas. I particularly like the approach of calculating the opponent's goals retrospectively via Equation 6 or 7, depending on the setting. This strikes me as an intuitive idea; if we're expecting the opponent to behave adversarially, we can assume that they were striving to reach the goal that would have hindered us the most. I also like the clean formulation of what a "subgoal" is, where it's simply a desired future state. This avoids the need for human experts to define the set of possible subgoals, which is a major limitation in a lot of other work that leverages the concept of subgoals (e.g., hierarchical RL). I agree with the overall motivation too; intuitively, it does seem better to predict subgoals rather than actions.

I'm not an expert in opponent modelling, but the approach seems fairly novel to me. While the approach does not yield a huge advantage in the experiments, it does achieve a moderate but persistent edge, and this is understandable since opponent-predicting ability is not the only driver of performance. Some of the side results are a worthy contribution too, e.g., the nice analysis in Figure 7(b). While I have some issues with the presentation (explained below), the quality of the writing in the introduction and related work section is excellent.

**Weaknesses:**

My main issue with the work is that I found the Method section from 3.2 onwards to be very hard to follow. I had to read it maybe 5 times before it started to make sense, and even now I'm not fully sure that I understand all the details. For example:
- One of the main points of confusion I have is that I'm not sure what $f_\psi$ and $p_\psi$ are supposed to be. They share the same parameters, $\psi$, so are they the same function, related functions, or is the notation accidentally overloaded?
- I don't follow how $p_\psi(\bar{g}|s^g)$ can be pre-trained "with the prior subgoal state $s_g$ being derived from the subgoal selector". Doesn't the subgoal selector leverage the value function? How do we already know the value function during the pre-training phase?
- How are goals represented to the Q-function? Are they encoded via $f_\psi$?
- I'm assuming that the training update at line 15 in Algorithm 1 can only be performed using the experience at time $t$ after $\bar{g}$ has later been inferred at time $t+H$. Is this correct? If so then it should be spelled out -- I got very confused at first trying to figure out how $\bar{g}$ could be known when performing the update.

Apologies if some of these questions don't make sense. I must admit that I'm not deeply familiar with variational inference, so my confusion might just reflect my own deficiencies. I'll be less concerned if the other reviewers found the methodology clear.

Beyond the above points, there are a few little issues:
- In the first paragraph of 3.3, "which as the policy's input" should presumably be "which *act* as the policy's input".
- The second sentence in the subsection "Subgoal inference model" ends weirdly.
- "You can tell a lot about a book from its cover" is a slightly strange analogy to use; I'd cut this.

While my overall confidence is low, my feeling is that the paper is not quite publishable in its current state. If the other reviewers found Sections 3.2 & 3.3 easy to understand then I'll be inclined to increase my score, but otherwise I think the explanation of the methodology needs to be improved a lot.

**Questions:**

Have I understood the general gist of the methodology correctly? If not, I'd really appreciate a dumbed down explanation, if that's possible.

---

> ### Author Response · Authors · 2023-11-23
> **Response to Reviewer ZFnJ**
>
> Thank you for acknowledging our novel contributions as well as raising valuable questions. Regarding the minor issues you raised, we have addressed them in the revision. Below is our detailed explanation of your concerns.
>
> > One of the main points of confusion I have is that I'm not sure what $f_\psi$ and $p_\psi$ are supposed to be. They share the same parameters, $\psi$, so are they the same function, related functions, or is the notation accidentally overloaded?
> > How are goals represented to the Q-function? Are they encoded via $f_\psi$?
>
> In essence, $f_\psi$ and $p_\psi$ are the same functions. The subgoal probability uses a normal distribution, with $p_{\psi}(\bar{g}|s^g)$ describes the probability distribution of subgoals. The mapping from state $s^g$ to subgoal $\bar{g}$ is expressed as $\bar{g} = f_\psi(s^g)$. This process involves encoding the state $s^g$ into the distribution and sampling from the distribution to derive $\bar{g}$. Apologies for the confusion, we makes it clearer in the revision.
>
> Yes, the subgoals encoded via $f_\psi$, and concatenate with the state as the input of Q-function.
>
> > I don't follow how $p_\psi(\bar{g}|s^g)$ can be pre-trained "with the prior subgoal state $s_g$ being derived from the subgoal selector". Doesn't the subgoal selector leverage the value function? How do we already know the value function during the pre-training phase?
>
> $p_\psi(\bar{g}|s^g)$ represents a pre-trained variational autoencoder.
>
> In the pre-training phase, the value function is unnecessary. We utilize data, namely states collected from any policy, to train the encoder capability of $p_\psi$. This phase does not involve a subgoal selector, rendering the need for a value function unnecessary.
>
> In the training phase, the parameters of $p_\psi$ is fixed, and the prior subgoal state $s^g$ derived from the subgoal selector is encoded as the subgoal $\bar{g}$.
>
> > I'm assuming that the training update at line 15 in Algorithm 1 can only be performed using the experience at time $t$ after $\bar{g}$ has later been inferred at time $t+H$. Is this correct? If so then it should be spelled out -- I got very confused at first trying to figure out how $\bar{g}$ could be known when performing the update.
>
> There are some misunderstandings here. OMG is an episodic off-policy method and the training update phase is performed on the replay buffer $\mathcal{D}$. That is to say, lines 6-10 in Algorithm 1 is executed repeatedly to collect the trajectories. Then, lines 11-15 in Algorithm 1 represent that OMG is trained on the replay buffer $\mathcal{D}$, where the state of the trajectory at timestep $t+H$ can be get to figure out the $\bar{g}$ at timestep $t$.

---

> ### Comment · Reviewer_ZFnJ · 2023-11-23
>
> Thanks for your responses. I think I need time to process the above, but some aspects make more sense to me now.
>
> > There are some misunderstandings here. OMG is an episodic off-policy method and the training update phase is performed on the replay buffer $\mathcal{D}$.
>
> Oh right, this is kind of obvious in hindsight.
>
> I think I understand why I was confused about $p_\phi$ now... It's not *predicting* subgoals, it's just a state embedding that's used to encode goal states compactly for the Q-function. I think the "g" notation confused me, because during the pre-training phase, the states being fed to the autoencoder aren't really "goals"; they're just arbitrary states to be encoded. Do correct me if I'm wrong... If I'm correct, I'd suggest altering the notation in the paper, since I don't think I was the only reviewer confused by this.

---

> ### Author Response · Authors · 2023-11-23
> **Response to Reviewer ZFnJ**
>
> > I think I understand why I was confused about $p_\phi$ now... It's not predicting subgoals, it's just a state embedding that's used to encode goal states compactly for the Q-function. I think the "g" notation confused me, because during the pre-training phase, the states being fed to the autoencoder aren't really "goals"; they're just arbitrary states to be encoded. Do correct me if I'm wrong... If I'm correct, I'd suggest altering the notation in the paper, since I don't think I was the only reviewer confused by this.
>
> OMG consists of three phases: pre-training, training and execution. In the pre-training phase, an autoencoder $p_{\psi}$ is trained with arbitrary states. In the training phase, an inference model $p_{\phi}$ is trained with prior subgoals encoded by $p_{\psi}$, which act as hindsight samples. Note that all the notations $g$ in the text refer to the subgoals in the training stage, not in the pre-training stage. Therefore, the states fed to the autoencoder $p_{\psi}$ are on the trajectory and are also the future states. Hence, we use the term "goal" to denote the embedding of the future state. In the execution phase, $p_{\phi}$ *predicts* the subgoal given the history trajectory without hindsight. I hope this solves your confusion.

---

> > ### Comment · Reviewer_ZFnJ · 2023-11-23
> >
> > Thanks for bearing with me -- yes, that does resolve my confusion. I've increased my score by 1 accordingly. I guess the chances of this paper getting up might be slim based on the other reviews, but I personally think it's a very interesting approach (particularly the minimax/maximax approach to deriving the opponent goals in hindsight). Since reviewers nLrr and 8YJ7 seem to be unclear on similar points to what I was, I hope you'll take the feedback regarding clarity on board, because this is good work that ought to be published eventually.

---

### Official Review · Reviewer_nLrr · 2023-11-02

**Soundness:** 3 good
**Presentation:** 2 fair
**Contribution:** 3 good
**Rating:** 3
**Confidence:** 4

**Summary:**

This paper presents a multiagent reinforcement learning algorithm that leverages opponent modelling to improve training performance (both sample efficiency and final performance), as well as generalization to partner strategies that were not seen during training.  They present a varian of "opponent modelling" that, rather than predicting the individual actions of other agents, predicts the state the joint strategy of all agents will likely reach within the next several steps (which they describe as a "subgoal").  The main goal of opponent modelling in this context is to compensate for the non-stationarity of the policies of other agents during training.  The intuition behind this work is that while an accurate one-step opponent model provides stationarity over a single step, a model of long-term outcomes provides approximate stationarity over several time steps.

Their method does not predict complete states, but learns a feature embedding as part of a generative model during a pretraining phase.  Salient subgoals are selected in hindsight from previous trajectories using a value-based heuristic, and this is used to train a recurrent network to predict future goals.  The RL agent's value function then conditions on the predicted subgoal in addition to the current state.  They present experimental results demonstrating that their method provides a modest performance improvement over independent Q-learning, as well as two existing opponent modelling methods, in several benchmark MARL environments (including the predator-prey particle environment, and the Starcraft multiagent challenge).

**Strengths:**

I felt that the core idea of the work has a lot of potential, and there is room for subsequent work on this idea.  Intuitively, accurate predictions of an opponent's behavior over a single timestep will be less informative than predictions of their long term behavior, particularly in settings where interactions between agents are in some sense "sparse".  While the empirical results are not definitive, they do support the claim that subgoal inference can improve over "naive" opponent modelling.

**Weaknesses:**

The main weakness of the work is the presentation, particularly the description of the OMG framework itself.  While it is implied that the parameters of the subgoal feature mapping (denoted as $\psi$) are trained using a VAE reconstruction loss on an initial batch of opponent data, the details of this process do not seem to have been provided.  As the subgoal representation encoded by $\psi$ would seem to play a large role in success or failure of the algorithm, the loss which $\psi$ minimizes should be provided.  A few other points that were unclear:
1. What form does the subgoal prior $p_{\psi}$ take, and what is its relation to the feature mapping $f_{\psi}$ (my guess is that $p_{\psi}$ is a normal distribution centered at $f_{\psi}$)
2. Similarly, what form does the posterior model $q_{\phi}$ take
3. How is the posterior goal prediction $\hat{g}$ derived from $q_{\phi}$?  Is it sampled from $q_{\phi}$, or do you choose $\hat{g}$ maximizing the likelihood under $q_{\phi}$?
4. It isn't completely clear whether the subgoal selection process used during the training of $\phi$ and $\theta$ is the same as that used during policy execution (either equation 6 or 7).
5. A minor point, but dropping the subscript in equation 5 makes it a little unclear that $\tau$ is the trajectory $\tau_t$ up to time $t-1$.
A less significant weakness is that the experimental results could be more comprehensive.  One concern is that OMG is only compared against IQL and other opponent modelling methods.  It would be helpful to see how well these perform when compared against centralized multi-agent RL methods such as QMIX.  It was also a little surprising that learning curves were not provided for the SMAC environments.

**Questions:**

1. During training, were all agents implementing OMG simultaneously?  I assume this was the case, but it was not made explicit.
2. What policies did agents follow during the pre-training phase (data collection for training $\psi$)?
3. How consistent are subgoal predictions over time?  Do they tend to remain stable over 2-3 timesteps?
4. Do you have any results where the VAE used to predict subgoals was trained without being conditioned on a pre-selected subgoal?  It would seem possible that you could get a similar result by simply conditioning Q-functions on a latent representation of opponent trajectories.
5. Could the sub-goal predictor $q_{\psi}$ have been trained with supervised learning on the outputs of the sub-goal selector?

---

> ### Comment · Reviewer_nLrr · 2023-11-22
> **Follow up to review**
>
> As the authors have not addressed the points of uncertainty regarding their method, I have reduced my score to reflect the unclear presentation.  The new score also reflects the fact that the authors have not addressed the connections to previous work on opponent modelling brought up by reviewer W6q1.

---

> > ### Author Response · Authors · 2023-11-23
> > **Response to Reviewer nLrr（Part Ⅰ）**
> >
> > Thanks for your valuable comments. Sorry, because of the problem of collecting experimental data, your comments was not addressed in time. As follows, we address your concerns in detail.
> >
> > > The main weakness of the work is the presentation, particularly the description of the OMG framework itself. While it is implied that the parameters of the subgoal feature mapping (denoted as $\psi$) are trained using a VAE reconstruction loss on an initial batch of opponent data, the details of this process do not seem to have been provided. As the subgoal representation encoded by $\psi$ would seem to play a large role in success or failure of the algorithm, the loss which $\psi$ minimizes should be provided.
> >
> > The subgoal's prior model $p_{\psi}(\bar{g}|s^g)$ is a VAE that learns from a set of states that are collected while training opponents. The optimization objective of VAE is :
> >
> > $<\hat{\omega}, \hat{\psi}> = \mathop{argmax}\limits_{\omega, \psi} \mathop{\mathbb{E}}\limits_{g \sim q_{\psi}(g|s)}\Big[\log p_{\omega}(s|g)\Big] - \mathrm{KL}\Big( q_{\psi}(g|s)||\mathcal{N}(0,1)\Big).$
> >
> > where $\psi$ and $\omega$ are parameters of the encoder and the decoder, respectively. The decoder $p_{\omega}(s|g)$, also denoted by $f^{-1}_{\psi}$, is also used to reconstruct the subgoal state, as shown in Section 4.5.
> > We have added details about $\psi$ in the appendix of revision.
> >
> > > What form does the subgoal prior $p_{\psi}$ take, and what is its relation to the feature mapping $f_{\psi}$ (my guess is that $p_{\psi}$ is a normal distribution centered at $f_{\psi}$). Similarly, what form does the posterior model $q_{\phi}$ take? How is the posterior goal prediction $\hat{g}$ derived from $q_{\phi}$? Is it sampled from $q_{\phi}$, or do you choose $\hat{g}$ maximizing the likelihood under $q_{\phi}$?
> >
> > The distribution of subgoal prior $p_{\psi}$ and subgoal posterior probability $q_{\phi}$ are used normal distribution. The prior subgoal $\bar{g}$ and posterior subgoal prediction $\hat{g}$ are sampled from $p_{\psi}$ and $q_{\phi}$ using reparameterization trick respectively, which is used to feed in policy net. The KL term in Eq(5) calculated using $\mu$ and $\sigma$ of normal distribution $p_{\psi}$ and $q_{\phi}$.
> >
> > > It isn't completely clear whether the subgoal selection process used during the training of $\phi$ and $\theta$ is the same as that used during policy execution (either equation 6 or 7).
> >
> > There is no subgoal selection process during execution phase because the state at timestep $t+k$ cannot be obtained at timestep $t$. The subgoal selection process exists only during training phase and uses the future state on the trajectories as prior about the subgoal.
> >
> > > A minor point, but dropping the subscript in equation 5 makes it a little unclear that $\tau$ is the trajectory $\tau_t$ up to time $t-1$.
> >
> > We rewrite the equation 5 in the revision to make it clearer, as follows:
> >
> > $<\hat{\theta}, \hat{\phi}> = \mathop{argmax}\limits_{\theta, \phi} \mathop{\mathbb{E}}\limits_{g \sim q_{\phi}(\hat{g}_t|\tau_t, s_t)}\Big[\log p_{\theta}(s_t|\hat{g}_t, \tau_t)\Big] - \mathrm{KL}\Big(q_{\phi}(\hat{g}_t|\tau_t, s_t)||p_{\psi}(\bar{g}_t|s^g)\Big)$
> >
> > where $\tau_t = \{s_0, a_0, a_0^o, \dots, s_{t-1}, a_{t-1}, a_{t-1}^{o}\}$
> >
> > >A less significant weakness is that the experimental results could be more comprehensive. One concern is that OMG is only compared against IQL and other opponent modelling methods. It would be helpful to see how well these perform when compared against centralized multi-agent RL methods such as QMIX. It was also a little surprising that learning curves were not provided for the SMAC environments.
> >
> > We have added the learning curves during SMAC to the appendix of the revision. We focus on autonomous agent that use opponent modeling to assist decision making, rather than depending on a centralized critic. Therefore, we do not compare our method with centralized multi-agent RL such as QMIX, as this would be an unfair comparison. In addition, QMIX has shown good performance on SMAC.
> >
> > > During training, were all agents implementing OMG simultaneously? I assume this was the case, but it was not made explicit. What policies did agents follow during the pre-training phase (data collection for training $\psi$)?
> >
> > During training, only the OMG agent's policy is trained, and the policies of the opponents are fixed. At the onset of each episode, the opponent’s policy is selected randomly from a set of pre-trained policies. The autonomous agent with OMG and the selected opponent interact in the environment. The autonomous agent’s index is also randomly determined. For Foraging, Predator-Prey, and SMAC environments, D3QN, PPO, and QMIX are used to pre-train the opponents respectively, while the states are collected for training the variational autoencoder $\psi$.
> >
> > These details about training are stated in the Appendix A.2, which we will make clearer in the revision.

---

> > ### Author Response · Authors · 2023-11-23
> > **Response to Reviewer nLrr（Part Ⅱ）**
> >
> > > A minor point, but dropping the subscript in equation 5 makes it a little unclear that $\tau$ is the trajectory $\tau_t$ up to time $t-1$.
> >
> > We rewrite the equation 5 in the revision to make it clearer, as follows:
> >
> > $<\hat{\theta}, \hat{\phi}> = \mathop{argmax}\limits_{\theta, \phi} \mathop{\mathbb{E}}\limits_{g \sim q_{\phi}(\hat{g}_t|\tau_t, s_t)}\Big[\log p_{\theta}(s_t|\hat{g}_t, \tau_t)\Big] - \mathrm{KL}\Big(q_{\phi}(\hat{g}_t|\tau_t, s_t)||p_{\psi}(\bar{g}_t|s^g)\Big)$
> >
> > where $\tau_t = (s_0, a_0, a_0^o, \dots, s_{t-1}, a_{t-1}, a_{t-1}^{o})$
> >
> > > How consistent are subgoal predictions over time? Do they tend to remain stable over 2-3 timesteps?
> >
> > Subgoal predictions $\hat{g}$ are made to assist OMG decision making and are made at every timestep. During training, we perform subgoal selection to choose $\bar{g}$, and it usually lasts a few steps. We used 100 trajectories and counted the selection frequency within the trajectory, as shown in the following table:
> >
> > | $s$ selection frequency | 1 | 2 | 3 | 4 | 5 |
> > | ------ | ------ | ------ | ------ | ------ | ------ |
> > | percentage | 35.4% | 20.0% | 22.8% | 19.3% | 2.5% |
> >
> > > Do you have any results where the VAE used to predict subgoals was trained without being conditioned on a pre-selected subgoal? It would seem possible that you could get a similar result by simply conditioning Q-functions on a latent representation of opponent trajectories.
> >
> > The method has outlined in [1], denoted as LIAM-VAE, involves conditioning Q-functions on a latent representation of opponent trajectories. However, comparative performance evaluations indicate that LIAM-VAE underperforms in comparison to LIAM. Consequently, we establish LIAM as the baseline algorithm due to its demonstrative capabilities.
> >
> > [1] Agent modelling under partial observability for deep reinforcement learning. NeurIPS, 2021.

---

### Meta-Review · Area_Chair_2KCF · 2023-12-06

**Metareview:**

Reviewers identified issues with clarity and novelty of this manuscript.

**Justification For Why Not Higher Score:**

Reviewers seemed quite confused by the presentation, it seems likely that critical details were missing.

**Justification For Why Not Lower Score:**

N/A

---

### Decision · Program_Chairs · 2024-01-16

Reject